# CASSI: Contextual and Semantic Structure-based Interpolation Augmentation for Low-Resource NER

**Tanmay Surana, Ho Thi Nga, Kyaw Zin Tun, Eng Siong Chng**

Nanyang Technological University, Singapore

tanmay003@e.ntu.edu.sg, {ngaht, ztkyaw, aseschng}@ntu.edu.sg

## Abstract

While text augmentation methods have been successful in improving performance in the low-resource setting, they suffer from annotation corruption for a token-level task like NER. Moreover, existing methods cannot reliably add context diversity to the dataset, which has been shown to be crucial for low-resource NER. In this work, we propose Contextual and Semantic Structure-based Interpolation (CASSI), a novel augmentation scheme that generates high-quality contextually diverse augmentations while avoiding annotation corruption by structurally combining a pair of semantically similar sentences to generate a new sentence while maintaining semantic correctness and fluency. To accomplish this, we generate candidate augmentations by performing multiple dependency parsing-based exchanges in a pair of semantically similar sentences that are filtered via scoring with a pretrained Masked Language Model and a metric to promote specificity. Experiments show that CASSI consistently outperforms existing methods at multiple low resource levels, in multiple languages, and for noisy and clean text.[1]

## 1 Introduction

Named Entity Recognition (NER) is the token-level task of classifying terms in text that belong to pre-defined entities like 'location', 'person', 'organization', etc. It is an important task in NLP that is used in recommendation systems, search and content classification systems for healthcare, academia, news organizations, etc., and as an integral part of other tasks like text summarization (Nallapati et al., 2016), question answering (Zhou et al., 2018; Fabbri et al., 2020), topic modeling (Krasnashchok and Jouili, 2018), etc. Annotating data for a token-level task like NER can be expensive, often requiring domain expertise. Therefore, NER datasets tend to be small, especially for narrow domains and low-resource languages.

Text Augmentation has been a successful method for improving performance in the low-resource setting for various tasks like Natural Language Inference, Sentiment Analysis, etc. It involves creating additional text using the existing training text while generally preserving the labels[2]. Typical methods for text augmentation include random distortions such as random deletions, insertions, substitutions, (Wei and Zou, 2019), word replacement (Wu et al., 2019; Kobayashi, 2018; Kim et al., 2022), backtranslation (Sennrich et al., 2016; Yu et al., 2018; Coulombe, 2018), paraphrasing (Madnani and Dorr, 2010; Kumar et al., 2019), and label-conditioned text generation through Language Models (Anaby-Tavor et al., 2020; Yoo et al., 2021), GANs (Sun and He, 2020), VAEs (Qiu et al., 2020), etc.

Preserving the token-level entity tags in NER is a challenging task and text augmentation methods often cause annotation corruption, i.e., assignment of incorrect token-level tags. Dai and Adel 2020 avoid the annotation corruption problem by performing random replacements of the same entity type. However, this creates limited diversity in the dataset and generates noisy sentences since the entity replacements don't take sentence context into account. Methods like backtranslation and paraphrasing show limited improvements and require an additional retagging step, which tends to be erroneous (Sharma et al., 2022; Kyaw, 2022). More effective works attempt to solve the label-preservation problem for low-resource NER by performing augmentations using a label-conditioned language model. The LM is conditioned on the tags by explicitly including the tags in the training

---

[1]Our code is available at https://github.com/tanmaysurana/CASSI

[2]Not all text augmentation methods necessarily preserve labels. For example, Yoon et al. 2021 and Harel-Canada et al. 2022.

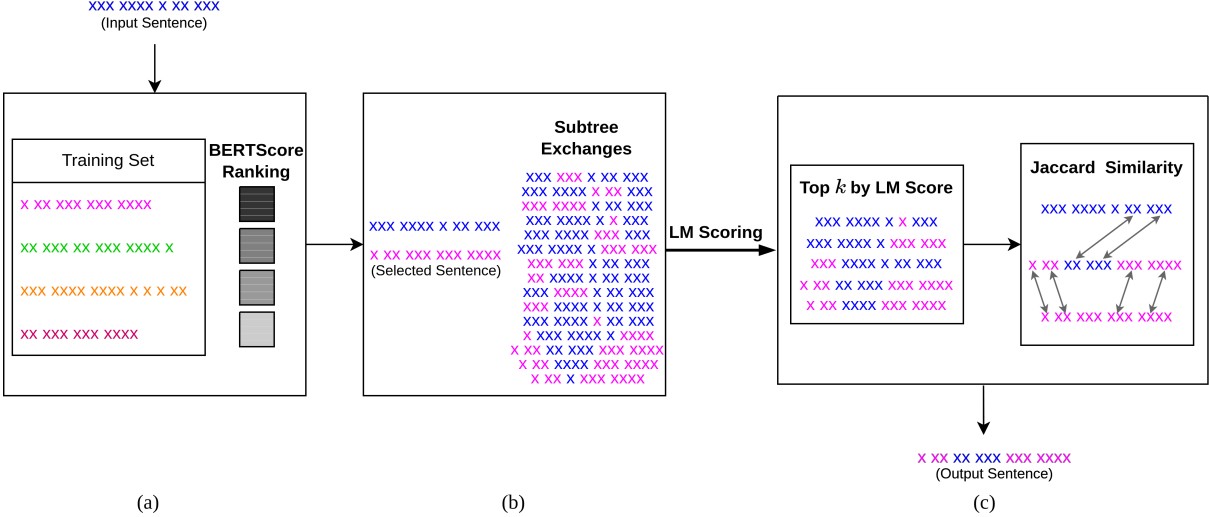

Figure 1: The Overall Pipeline of Our Method. Best viewed in color.

sentences, in a process called label linearization. DAGA (Ding et al., 2020) uses this process to generate new sentences including entity tags. However, these sentences tend to be ungrammatical and disfluent in nature when trained on a small number of examples. MELM (Zhou et al., 2022) uses this process to perform entity replacements through masked generation using a pretrained Masked Language Model, such that the replacements take sentence context into account, but does not introduce new context patterns into the dataset. Both of these methods are also able to add new entities to the training set. These methods outperform simpler methods like random substitution, random deletion, etc. However, these methods involve training a language model on limited data and are therefore still prone to annotation corruption, generating noisy training examples. This also severely limits the number of augmentations that can be added.

Moreover, Lin et al. 2020 show that, in the low-resource setting, decreasing contextual diversity in the training set more significantly impacts NER performance than reducing entity diversity. Therefore, contextual diversity is more important than entity regularity knowledge to generalize to unseen examples in the low-resource setting. While transformer language models are pretrained on large corpora, they are not able to sufficiently capture context patterns for NER when fine-tuned on a small number of examples. Contextual augmentation has been applied to the token-level task of Aspect Term Extraction by Li et al. 2020, who finetune MASS (Song et al., 2019) to generate new partial contexts

for English text by replacing 'O' tags. However, this requires additional in-domain data. Moreover, MASS is only pretrained on 4 languages: English, German, French, and Romanian. Additionally, Shi et al. 2021 alter the content while maintaining sentence structure by performing random substitutions of same-label spans for few-shot structured prediction tasks where annotations contain structural information like part-of-speech tagging, dependency parsing, and constituency parsing.

In this work, we propose Contextual and Semantic Structure-based Interpolation (CASSI) as a text augmentation scheme for low-resource NER. With the goal of increasing contextual diversity and avoiding annotation corruption, we use dependency parse trees to identify subtrees containing the subject, object, or complement between pairs of semantically similar sentences. We then generate candidate augmentations by performing exchanges between all pairs of subject subtrees and all pairs of object/complement subtrees between the two sentences. To ensure semantic correctness and fluency, we filter the resulting candidates by scoring them with a language model, similar to rescoring methods used in Automatic Speech Recognition (Xu et al., 2018; Chan et al., 2016) and Neural Machine Translation (Gülçehre et al., 2015). We leverage an existing pretrained Masked Language Model to avoid training a language model on limited data. Furthermore, we take steps to reduce the bias from the language model against specific names and towards shorter sentences and high-frequency generic terms. Specifically, we pick the top $k$ scored sen-

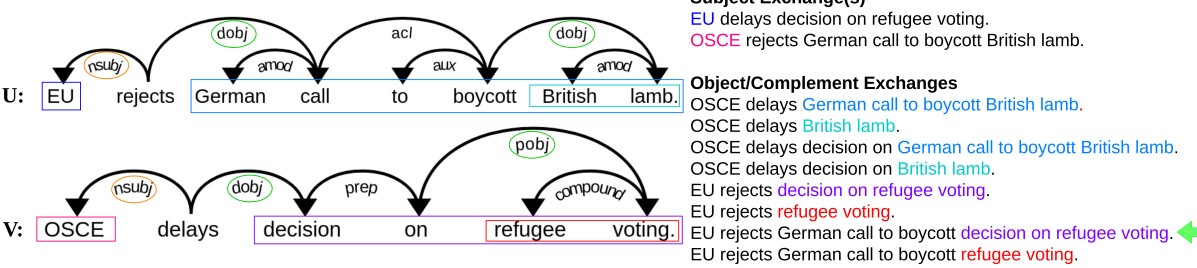

Figure 2: An example of subtree exchanges to generate candidate augmentations. Every colored box is a subject/object/complement subtree. The green arrow indicates the candidate chosen as the final augmentation by our method. Best viewed in color.

tences and further filter them using a simple metric based on Jaccard scores between the candidate augmentations and the original sentences.

Our contributions can be summarized as follows: (1) we present a novel text augmentation scheme that improves context diversity in the dataset while avoiding annotation corruption, (2) we show that our method consistently and significantly outperforms existing methods in multiple languages for monolingual and cross-lingual NER, and (3) we show that our method outperforms existing methods on noisy social media text.

## 2 Proposed Method

Our proposed method (shown in Figure 1) generates augmentations by combining a pair of semantically similar sentences to generate a new sentence while ensuring a high degree of fluency, and contextual and semantic correctness. For every sentence in the training set, we first rank the remaining sentences by sentence similarity (Figure 1a; Section 2.1). Next, we generate multiple candidate augmentations by performing structure-based exchanges between the given sentence and a selected semantically similar sentence using their dependency parse trees (Figure 1b; Section 2.2). We then filter out a subset of the candidate augmentations by scoring them with a pretrained Masked Language Model. We select the final augmentation from this subset using a simple metric to promote specificity and diversity (Figure 1c; Section 2.3).

### 2.1 Sentence Selection

Since sentence semantics and context clues are important for detecting entities, for a given sentence in the training set we rank the remaining sentences using a sentence similarity metric. Therefore, to generate $n$ augmentations, we use the top $n$ sentences according to their similarity to the given

sentence.

We use BERTScore (Zhang et al., 2020) as the sentence similarity metric. BERTScore is a token-wise metric used to evaluate text generation for tasks like machine translation and image captioning. It calculates Precision (P) and Recall (R) by matching tokens in the source and target sentences by cosine similarities between their embeddings retrieved from an MLM like BERT (Devlin et al., 2019). For a source sentence $\mathbf{A}$ with normalized embeddings $\{a_1, ..., a_p\}$ and a target sentence $\mathbf{B}$ with normalized embeddings $\{b_1, ..., b_q\}$:

$$P = \frac{1}{p} \sum_{a_i \in \mathbf{A}} \max_{b_j \in \mathbf{B}} a_i^\top b_j; R = \frac{1}{q} \sum_{b_j \in \mathbf{B}} \max_{a_i \in \mathbf{A}} a_i^\top b_j$$

The final score is the F1 score ($F1 = 2\frac{PR}{P+R}$). We choose BERTScore because it is able to use existing MLMs directly, without additional modifications like calculating sentence-level embeddings (Reimers and Gurevych, 2019). Moreover, we expect the token-wise nature of BERTScore to make it more robust to noisy social media text for a token-level task like NER.

### 2.2 Generating Candidate Augmentations

Figure 2 shows an example of subtree exchanges to generate candidate augmentations. For a given sentence $\mathbf{U}$ and a selected sentence $\mathbf{V}$, we use dependency parsing to identify *subject subtrees* and *object/complement subtrees*. A subject subtree is the subtree of a verb that contains the verb's subject, and an object/complement subtree is the subtree of a verb or preposition that contains the verb or preposition's object or complement. To generate candidate augmentations, we perform exchanges within all possible pairs of subject subtrees between $\mathbf{U}$ and $\mathbf{V}$. Similarly, we perform exchanges within

all possible pairs of object/complement subtrees between **U** and **V**. Therefore, each such exchange generates a pair of candidate augmentations.

We choose subjects, objects, and complements for exchanges as they are likely to contain entities and context clues for entities. We use subtrees as the structures to exchange as they contain more information than just the subject, object, or complement or the noun chunks containing them, thus generating more diverse candidate augmentations.

Since we consider all exchanges between **U** and **V** as candidate augmentations, we avoid selecting duplicate pairs of sentences. Therefore, if **V** is selected for **U**, **U** is skipped when selecting sentences for **V**. For sentences where the dependency parse cannot find at least one subject or one object, we replace entities in the sentence with random entities of the same type to avoid simply oversampling these sentences.

## 2.3 Filtering Candidates

To select the final augmentation, we filter the candidate augmentations in two steps: (1) scoring with a Language Model, and (2) filtering for specificity and diversity. We select just one augmentation for every pair of sentences to increase the diversity of the augmentations.

### 2.3.1 Language Model Scoring

Language Model scoring is used in Automatic Speech Recognition and Machine Translation to ascertain the correctness of a sentence (transcription or translation) by calculating the probability of occurrence of each token in the sentence. We use LM scoring to score the candidate augmentations for every pair of sentences. We calculate the LM score using a pretrained MLM following Salazar et al. 2020. The score is calculated as the sum of the log-likelihoods of each token in the sentence given all other tokens. Also following Salazar et al. 2020, we normalize the score by the number of *words* in the sentence (ignoring punctuation) to counter the effects of sentence length on the score. Formally, given a sentence **S** with tokens $\{t_1, ..., t_m\}$, the LM score is calculated as:

$$\text{LMScore}(\mathbf{S}) = \frac{1}{M} \sum_{i=1}^{m} \log P(t_i \mid \mathbf{S} - \{t_i\}; \Theta)$$

where $M$ is the number of words in the sentence, and $\Theta$ represents the MLM's parameters.

| Sentence | LM Score |
|---|---|
| I will be staying at the hotel | -0.74 |
| I will be staying at the Hilton | -1.29 |
| John will be staying at the hotel | -1.75 |
| John Doe will be staying at the hotel | -2.19 |

Table 1: Example of LM bias against specificity. The LM score decreases for more specific sentences. LM scores are pseudo-log-likelihoods calculated using XLM-R$_{\text{BASE}}$ following Salazar et al. 2020. The scores are normalized by the number of words in the sentence.

### 2.3.2 Filtering for Specificity and Diversity

As shown in Table 1, LM scoring is inherently biased against specificity. Generic terms are assigned higher probabilities than specific names since they occur more frequently, which is detrimental to a task like NER. To reduce this bias and promote diversity in the augmentations, we pick the top $k$ LM-scored candidates and further filter them with the geometric mean of the Jaccard similarities between the words in the candidates and the words in the original pair of sentences. Given a sentence **U**, a sentence **V**, and a candidate augmentation **C** as *sets* of words:

$$\text{J-Score}(\mathbf{U}, \mathbf{V}, \mathbf{C}) = \sqrt{\frac{|\mathbf{U} \cap \mathbf{C}||\mathbf{V} \cap \mathbf{C}|}{|\mathbf{U} \cup \mathbf{C}||\mathbf{V} \cup \mathbf{C}|}}$$

The selected augmentation is therefore biased towards containing more words from both original sentences, thus increasing the overall specificity of the augmentation. For the same reason, the selected augmentation is likely to be more diverse than the other $k - 1$ candidates (since the selected augmentation is biased against being disproportionately more similar to one of the two original sentences).

## 2.4 Post-Processing

In the event that a multi-word entity gets split by dependency parsing (for example, only the last name is picked from a name containing both first and last names), the entity's tag is fixed according to the IOB scheme to ensure that all entities (1) start with a B- tag, (2) end with an I- tag, and (3) only contain I- tags in between.

## 3 Experiments

We evaluate our augmentation scheme on low-resource monolingual and cross-lingual NER in multiple languages. We also evaluate our method

| # Gold | Method | En | De | Es | Nl | Avg |
|---|---|---|---|---|---|---|
| 100 | Gold Only | 57.11 / 41.92 | 39.52 / 30.40 | 57.93 / 49.75 | 41.58 / 33.51 | 49.03 / 38.89 |
| | DAGA | 60.30 / 56.52 | 52.11 / 45.21 | 60.82 / 56.41 | 45.44 / 40.06 | 54.67 / 49.55 |
| | MELM | 67.36 / 62.06 | 53.88 / 49.12 | 62.61 / 58.62 | 47.32 / 43.38 | 57.79 / 53.30 |
| | CASSI (*Ours*) | **75.04 / 70.04** | **65.96 / 58.54** | **69.93 / 65.35** | **64.09 / 58.13** | **68.75 / 63.02** |
| 250 | Gold Only | 75.64 / 70.52 | 64.37 / 58.19 | 69.38 / 65.15 | 60.84 / 56.94 | 67.56 / 62.70 |
| | DAGA | 74.87 / 69.89 | 59.85 / 54.16 | 68.75 / 64.75 | 59.80 / 57.23 | 65.82 / 61.51 |
| | MELM | 77.48 / 72.47 | 63.50 / 57.23 | 70.46 / 66.36 | 64.08 / 60.22 | 68.88 / 64.07 |
| | CASSI (*Ours*) | **82.02 / 77.08** | **72.79 / 65.26** | **75.89 / 71.79** | **72.90 / 68.23** | **75.90 / 70.59** |
| 500 | Gold Only | 79.98 / 75.02 | 68.69 / 62.42 | 72.91 / 68.81 | 67.18 / 64.90 | 72.19 / 67.79 |
| | DAGA | 80.60 / 75.51 | 71.42 / 65.48 | 76.43 / 72.64 | 69.51 / 66.34 | 74.49 / 69.99 |
| | MELM | 81.84 / 76.43 | 69.44 / 64.16 | 75.66 / 72.53 | 71.29 / 68.68 | 74.56 / 70.45 |
| | CASSI (*Ours*) | **85.03 / 80.24** | **76.69 / 70.47** | **79.60 / 76.42** | **77.17 / 74.13** | **79.62 / 75.31** |
| 750 | Gold Only | 83.69 / 78.82 | 75.07 / 69.17 | 76.88 / 73.54 | 74.42 / 71.16 | 77.52 / 73.17 |
| | DAGA | 84.35 / 79.58 | 73.81 / 68.45 | 77.90 / 74.57 | 72.09 / 69.96 | 77.04 / 73.14 |
| | MELM | 84.06 / 78.84 | 73.87 / 68.01 | 78.50 / 75.26 | 74.29 / 72.40 | 77.68 / 73.63 |
| | CASSI (*Ours*) | **86.39 / 81.83** | **77.88 / 72.19** | **80.51 / 77.62** | **78.82 / 75.77** | **80.90 / 76.85** |
| 1k | Gold Only | 85.50 / 80.86 | 77.00 / 71.82 | 78.55 / 75.78 | 75.88 / 73.26 | 79.23 / 75.43 |
| | DAGA | 85.21 / 80.62 | 76.64 / 70.98 | 79.72 / 76.90 | 75.97 / 74.23 | 79.38 / 75.68 |
| | MELM | 85.56 / 80.56 | 76.74 / 71.07 | 80.20 / 77.33 | 76.08 / 73.92 | 79.64 / 75.72 |
| | CASSI (*Ours*) | **87.24 / 82.86** | **79.48 / 73.71** | **81.25 / 78.82** | **80.08 / 76.90** | **82.01 / 78.07** |
| 2k | Gold Only | 88.42 / 83.94 | 80.42 / 75.00 | 82.77 / 80.26 | 80.16 / 77.85 | 82.94 / 79.26 |
| | DAGA | 88.53 / 84.27 | 79.45 / 74.08 | 82.70 / 80.16 | 79.97 / 77.68 | 82.66 / 79.05 |
| | MELM | 88.39 / 83.79 | 79.00 / 73.42 | 82.67 / 80.26 | 79.97 / 78.65 | 82.51 / 79.03 |
| | CASSI (*Ours*) | **88.68 / 84.49** | **80.78 / 75.20** | **83.06 / 80.66** | **82.55 / 79.83** | **83.77 / 80.05** |
| 4k | Gold Only | **90.01 / 85.90** | 82.32 / **77.40** | 84.60 / 82.23 | 81.39 / 79.62 | 84.58 / 81.29 |
| | DAGA | 89.82 / 85.70 | 82.00 / 76.96 | 84.23 / 81.85 | 77.94 / 77.31 | 83.50 / 80.45 |
| | MELM | 89.58 / 85.20 | 82.00 / 76.65 | 84.10 / 81.74 | 79.56 / 79.40 | 83.81 / 80.75 |
| | CASSI (*Ours*) | **90.02 / 85.97** | **82.45** / 77.20 | **84.97 / 82.75** | **83.51 / 81.10** | **85.24 / 81.75** |

Table 2: Results of Monolingual NER on subsets of CoNLL 2002/2003. Results are reported as Micro-F1 / Macro-F1. Numbers in bold indicate best performance

on noisy social media datasets. We compare our method against MELM (Zhou et al., 2022) and DAGA (Ding et al., 2020).

### 3.1 Datasets

For monolingual NER, we perform experiments on the CoNLL 2002/2003 (Tjong Kim Sang, 2002; Tjong Kim Sang and De Meulder, 2003) datasets in four languages: English (En), German (De), Spanish (Es), and Dutch (Nl). For each language, we generate subsets of the training set of size 100, 250, 500, 750, 1k, 2k, and 4k. To concretely study the effects of adding data, we sample the subsets progressively, i.e., each smaller subset is sampled from the next larger subset.

For zero-shot cross-lingual NER, we evaluate models trained using the subsets of the English training set and the standard English dev set on the German, Spanish, and Dutch test sets.

Additionally, we evaluate the performance of our method in conditions where dependency parsing is likely to be error-prone. For this, we perform experiments on the following social media datasets: WeiboNER (Peng and Dredze, 2015) - a Chinese social media dataset containing 1.4k sentences in the train-

ing set, WNUT-2017 (Derczynski et al., 2017) - an English tweets dataset containing 3.3k sentences in the training set, and ReLDI-NormTagNER-hr 2.0 (Miličević and Ljubešić, 2016) - a Croatian tweets dataset.

For all datasets except ReLDI-NormTagNER-hr 2.0, we use the standard dev and test sets. For ReLDI-NormTagNER-hr 2.0, we randomly split the dataset into 4.8k training sentences, 1.6k dev sentences, and 1.6k test sentences since the dataset source does not contain standard splits. All datasets follow the IOB annotation scheme.

### 3.2 Experimental Setting

#### 3.2.1 Augmentation

**Baseline Methods** For DAGA and MELM, we follow augmentation settings as described in Ding et al. 2020 and Zhou et al. 2022, respectively.

**CASSI** We use XLM-R$_{BASE}$ (Conneau et al., 2020) for both calculating the pairwise BERTScore and LM scoring. For dependency parsing, we use the following language-specific pipelines from SpaCy (Honnibal et al., 2020): en_core_web_trf for English, de_dep_news_trf

| # Gold | Method | En → De | En → Es | En → Nl | Avg |
|---|---|---|---|---|---|
| 100 | Gold Only | 40.65 / 29.13 | 38.60 / 30.34 | 35.04 / 25.70 | 38.10 / 28.39 |
| | DAGA | 38.61 / 34.18 | 39.27 / 35.88 | 36.96 / 36.68 | 38.28 / 35.58 |
| | MELM | 40.95 / 35.14 | 45.10 / 37.67 | 34.36 / 33.67 | 40.14 / 35.49 |
| | CASSI (*Ours*) | **54.83 / 42.83** | **54.94 / 45.42** | **54.65 / 48.68** | **54.81 / 45.64** |
| 250 | Gold Only | 47.77 / 39.30 | 50.91 / 45.06 | 45.24 / 43.79 | 47.97 / 42.72 |
| | DAGA | 50.99 / 43.92 | 51.95 / 45.53 | 49.89 / 44.65 | 50.94 / 44.70 |
| | MELM | 58.80 / 50.64 | 57.54 / 50.43 | 55.60 / 51.54 | 57.31 / 50.87 |
| | CASSI (*Ours*) | **64.90 / 53.37** | **61.65 / 53.25** | **63.63 / 58.13** | **63.39 / 54.92** |
| 500 | Gold Only | 51.93 / 42.67 | 54.52 / 47.66 | 50.81 / 48.75 | 52.42 / 46.36 |
| | DAGA | 60.71 / 52.24 | 59.95 / 53.37 | 57.98 / 53.15 | 59.55 / 52.92 |
| | MELM | 63.34 / 55.86 | 62.34 / 55.97 | 59.25 / 55.83 | 61.64 / 55.89 |
| | CASSI (*Ours*) | **67.18 / 56.57** | **64.45 / 56.50** | **61.84 / 58.28** | **64.49 / 57.12** |
| 750 | Gold Only | 63.53 / 52.70 | 60.61 / 53.78 | 61.24 / 57.76 | 61.79 / 54.75 |
| | DAGA | 66.09 / 56.12 | 62.97 / 56.06 | 62.03 / 57.79 | 63.70 / 56.66 |
| | MELM | 65.17 / 56.47 | 63.43 / 56.63 | 58.87 / 55.64 | 62.49 / 56.25 |
| | CASSI (*Ours*) | **68.60 / 58.57** | **64.77 / 57.79** | **63.09 / 59.51** | **65.49 / 58.62** |
| 1k | Gold Only | 66.02 / 56.60 | 63.44 / 57.02 | 60.67 / 58.05 | 63.38 / 57.22 |
| | DAGA | 66.21 / 58.32 | 65.11 / 58.29 | 62.20 / 59.36 | 64.51 / 58.66 |
| | MELM | 67.47 / 59.42 | 64.97 / 57.81 | 61.99 / 58.94 | 64.81 / 58.72 |
| | CASSI (*Ours*) | **69.64 / 60.15** | **66.68 / 59.22** | **66.90 / 63.47** | **67.74 / 60.95** |
| 2k | Gold Only | 70.72 / 62.36 | 68.02 / **62.01** | 68.19 / 65.78 | 68.98 / 63.38 |
| | DAGA | 71.52 / **62.66** | **68.40** / 61.45 | 67.71 / 64.84 | 69.37 / 62.98 |
| | MELM | 68.86 / 61.61 | 67.96 / 61.87 | 67.28 / 64.03 | 68.03 / 62.50 |
| | CASSI (*Ours*) | **71.54** / 62.52 | 68.22 / 61.59 | **68.74 / 66.76** | **69.50 / 63.62** |
| 4k | Gold Only | 72.01 / 63.65 | **70.00 / 64.02** | 68.30 / 66.23 | 70.10 / 64.63 |
| | DAGA | 71.95 / 63.98 | 69.44 / 63.25 | **69.86 / 67.69** | 70.42 / 64.97 |
| | MELM | 71.48 / 63.87 | 68.92 / 63.20 | 67.09 / 66.09 | 69.16 / 64.39 |
| | CASSI (*Ours*) | **72.47 / 64.29** | 69.83 / 63.97 | 69.25 / 67.50 | **70.52 / 65.25** |

Table 3: Results of Cross-lingual NER on subsets of CoNLL 2002/2003. Results are reported as Micro-F1 / Macro-F1. Numbers in bold indicate best performance. Underlined numbers indicate competitive performance.

for German, `es_dep_news_trf` for Spanish, `nl_core_news_lg` for Dutch, `zh_core_web_trf` for Chinese, and `hr_core_news_lg` for Croatian. For all experiments, we set the number of candidates selected for J-Score filtering ($k$) to 5 based on preliminary experiments.

### 3.2.2 NER Training

We perform all experiments using mBERT$_{BASE}$-uncased-BiLSTM-CRF (Vasantharajan et al., 2022a,b). We train with a batch size of 32 for 50 epochs using the AdamW optimizer. All training sets up to size 500 (including augmentations) are trained with a learning rate of $2e - 4$ with the number of warm-up steps set to 200. For training sets larger than 500 sentences, we decrease the learning rate and increase the number of warm-up steps proportionally with the increase in training set size. For the CoNLL subsets, we average the performance on three randomly sampled subsets of the same size. For the social media NER datasets, we average the performance on three runs. We report both Micro-F1 and Macro-F1 in each case.

For all augmentation methods, given a gold training set of size $N$, the number of augmentations we generate ranges from $N$ up to $10N$. For gold training sets of size 1k and larger, we also generate $0.25N$, $0.5N$, and $0.75N$ augmentations, by randomly sampling from the set of augmentations of size $N$. For each method, we stop adding augmentations if there is no improvement in the best dev set Micro-F1 for three consecutive increments in the number of augmentations. For this purpose, we track the average dev set Micro-F1 across the three runs, so as to report a single augmentation size in each case.

### 3.3 Results

**Monolingual NER** Table 2 summarizes the results on monolingual NER. Our method consistently outperforms the baseline methods for all languages. There is significant improvement across languages for low-resource levels up to 1k. The improvement becomes marginal for subset sizes of 2k and 4k for English, German, and Spanish. This is consistent with observations from Lin et al. 2020 that increasing context diversity in a clean text regular NER dataset has a limited effect on the performance for over 2k sentences when fine-tuning a pretrained transformer. However, the per-

| Method | WNUT-2017 | WeiboNER | ReLDI-Hr 2.0 |
|---|---|---|---|
| Gold Only | 41.81 / 36.01 | 62.49 / 53.78 | 72.58 / 52.45 |
| DAGA | 38.87 / 31.68 | 64.45 / 55.99 | 71.70 / 52.88 |
| MELM | 42.12 / 36.27 | 62.80 / 54.21 | 72.49 / 51.99 |
| CASSI (*Ours*) | **43.22 / 37.31** | **65.02 / 56.08** | **72.90 / 53.19** |

Table 4: Results of NER on Social Media Datasets. Results are reported as Micro-F1 / Macro-F1.

formance continues to significantly improve for Dutch. We discuss this further in Section 4.1. On average, MELM and DAGA marginally improve performance for 1k sentences but marginally reduce performance for 2k and 4k sentences. Given that performance improvements from augmentations diminish as dataset size increases, we attribute this to the noise in the augmentations outweighing any diversity introduced by them.

**Cross-lingual NER**   Table 3 summarizes the results on zero-shot cross-lingual NER with English as the source language. On average, our method outperforms both MELM and DAGA for all subset sizes. Interestingly, DAGA becomes competitive in the Micro-F1 for resource levels of 2k and 4k sentences. With enough examples, DAGA can produce better context patterns. While these hurt performance in the monolingual setting, they seem to help generalize to unseen entities in other languages. However, it more notably underperforms in the Macro-F1 compared to our method. We posit that this is because the auto-regressive generation of entity tags along with the sentences causes it to disproportionately generate more frequent tags. In the case of MELM, the lack of new context patterns and annotation corruption cause the performance to drop for 2k and 4k sentences.

**NER on Noisy Text**   Table 4 summarizes the results of our experiments on social media datasets. Our method consistently outperforms baseline methods even for noisy social media text where dependency parsing is likely to be erroneous. Moreover, these datasets tend to contain a considerable number of mislabeled examples and a stilted distribution of tags (for example, the 'person' tag occurs a lot more frequently than other tags in these datasets), which can make learning a reliable distribution of labels difficult for label-conditioning methods.

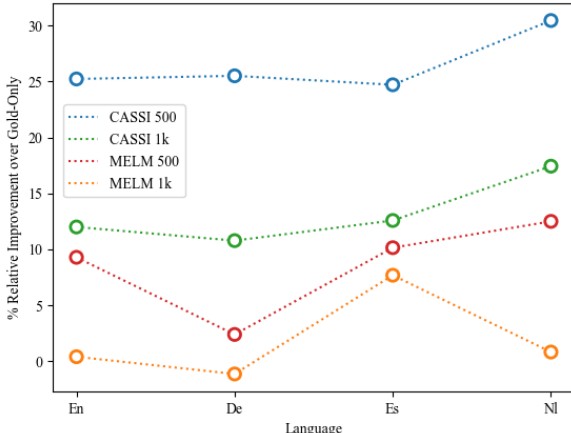

Figure 3: Relative Improvement (%) of Micro-F1 over Gold-Only, calculated as $(F1 - F1_{Gold})/(100 - F1_{Gold})$. Best viewed in color.

## 4   Discussion

### 4.1   Performance on Dutch

Figure 3 shows the language-wise relative improvement of CASSI and MELM over the Gold-Only baseline for 500 and 1k sentences. For CASSI, across both resource levels, the relative improvements for English, German, and Spanish are very similar whereas the improvement for Dutch is significantly higher. Given that the datasets are from the same domain (news articles), the most salient difference in the experiment settings for these languages is the amount of pretraining they received in mBERT. mBERT is pretrained on the entire Wikipedia dump of the top hundred languages with the largest Wikipedias[3]. The English, German, and Spanish Wikipedias contain 3.3B, 1.1B, and 800M words respectively, however, the Dutch Wikipedia is considerably smaller at 300M words[4]. We suspect that Dutch gains more from contextual diversity in the augmentations due to lower representation during pretraining. This is consistent with observations from Lin et al. 2020 who conclude that pretraining helps capture context patterns when finetuning for NER. This implies that context diversity in the augmentations is more important for low-resource languages with low representation during pretraining. Figure 3 also shows that MELM, an entity replacement method that does not introduce new context patterns, does not follow this pattern.

---

[3]https://github.com/google-research/bert/blob/master/multilingual.md

[4]As of March/April 2019. Collected using https://en.wikipedia.org/wiki/List_of_Wikipedias and https://archive.org/web

| Method | En 1k | De 1k | Es 1k | Nl 1k |
|---|---|---|---|---|
| Random Selection | **87.43** / 83.00 | 78.88 / 72.64 | 81.29 / 78.81 | 79.16 / 76.65 |
| BERTScore | 87.38 / **83.09** | **79.10** / **73.47** | **81.44** / **79.02** | **79.80** / **77.34** |

Table 5: BERTScore Ablation. Results are reported as Micro-F1 / Macro-F1.

| Method | En 1k | De 1k | Es 1k | Nl 1k |
|---|---|---|---|---|
| Best LM Score | 87.26 / 82.90 | **79.51** / **74.12** | 80.95 / 78.40 | 78.99 / 76.42 |
| Random (Top 5) | 86.88 / 82.70 | 79.46 / 74.08 | 80.73 / 78.09 | 79.05 / 76.72 |
| J-Score (Top 5) | **87.38** / **83.09** | 79.10 / 73.47 | **81.44** / **79.02** | **79.80** / **77.34** |

Table 6: J-Score Ablation. Results are reported as Micro-F1 / Macro-F1.

## 4.2 Ablations

We perform ablation studies on the 1k subsets of CoNLL 2002/2003 to evaluate the effect of (1) matching sentences by similarity, and (2) using the J-Score filter for specificity. To assess the quality of the augmentations, we compare the performance on the test set upon adding $10N$ augmentations (where $N$ = 1k). As mentioned in Section 3.2.2, for each language, we average the results on three sampled subsets of size 1k.

### 4.2.1 BERTScore

Table 5 compares using BERTScore to select semantically similar sentences with selecting sentences at random. On average, matching sentences by similarity outperforms random selection.

### 4.2.2 J-Score Filter

We compare the performance of selection from the top $k$ candidates using the J-Score metric for specificity against (1) selecting the candidate with the best LM score, and (2) randomly selecting a candidate from the top $k$. Table 6 shows these results. On average, filtering using the J-score metric outperforms the other methods. It notably outperforms random selection from top $k$, showing that using the J-score metric does indeed select sentences with useful properties.

## 4.3 Quality of Augmentations

Here, we consider the quality of our augmentations by (1) comparing the number of performance-improving augmentations added by each method, and (2) comparing the 'naturalness' (via LM scoring) of our augmentations to the gold sentences.

### 4.3.1 Number of Augmentations

Table 7 summarizes the number of augmentations added for each method on the CoNLL 2002/2003 subsets. We are able to add considerably more

| # Gold | Method | En | De | Es | Nl | Avg |
|---|---|---|---|---|---|---|
| 100 | DAGA | 7 | 8 | 8 | 9 | 8 |
| | MELM | 5 | 2 | 5 | 3 | 3.75 |
| | CASSI (*Ours*) | **10** | **9** | **10** | 8 | **9.25** |
| 250 | DAGA | 2 | 4 | 4 | 2 | 3 |
| | MELM | 4 | 1 | 7 | 3 | 3.75 |
| | CASSI (*Ours*) | **8** | **9** | 7 | **9** | **8.25** |
| 500 | DAGA | 1 | 1 | 3 | 1 | 1.5 |
| | MELM | 2 | 1 | 2 | 1 | 1.5 |
| | CASSI (*Ours*) | **9** | **5** | **7** | **4** | **6.25** |
| 750 | DAGA | 1 | 1 | 1 | 1 | 1 |
| | MELM | 2 | 1 | 2 | 1 | 1.5 |
| | CASSI (*Ours*) | **4** | **4** | **10** | **6** | **6** |
| 1k | DAGA | 0.5 | 0.25 | 1 | 0.75 | 0.63 |
| | MELM | 0.25 | 0.5 | 0.75 | 0.75 | 0.56 |
| | CASSI (*Ours*) | **5** | **1** | **8** | **6** | **5** |
| 2k | DAGA | 0.25 | 0.25 | 0.25 | 0.25 | 0.25 |
| | MELM | 0.25 | **0.75** | 0.5 | 0.25 | 0.44 |
| | CASSI (*Ours*) | **1** | 0.25 | **2** | **2** | **1.31** |
| 4k | DAGA | 0.25 | 0.25 | 1 | 0.25 | 0.44 |
| | MELM | 0.25 | 0.25 | 0.5 | 0.25 | 0.31 |
| | CASSI (*Ours*) | **0.75** | **1** | **2** | **2** | **1.44** |

Table 7: Number of augmentations ($\times N$, where $N$ is the number of Gold samples) added for each method. The higher quality of our augmentations allows us to add more augmentations than the baseline methods.

performance-improving augmentations than baseline methods. Note that we do not go beyond $10N$ augmentations in any case.

Additionally, we evaluated all methods on *all* augmentation increments up to $10N$ on the 500 sentence subsets of CoNLL 2002/2003 to compare the dev set Micro-F1 for every method. The results are summarized in Figure 4. Our method outperforms the baseline methods on all augmentation increments for all languages.

### 4.3.2 Naturalness

We performed LM scoring on the 1k subsets of CoNLL 2002/2003, and 1k augmentations generated from them. We scored the sentences using mBERT$_{BASE}$-uncased (we do not use XLM-R$_{BASE}$ since the augmentations were selected using XLM-R$_{BASE}$) following Salazar et al. 2020 and normalized them by the number of words (ignoring punctuation). The augmentations are scored separately, i.e., they are not added to the gold-only set. The LM Scores are pseudo-log-likelihoods, with greater values being better. We present the mean and standard deviation of the LM scores over the sets. Table 8 shows that the augmentations score very similarly (albeit marginally lower on average) to the gold sentences.

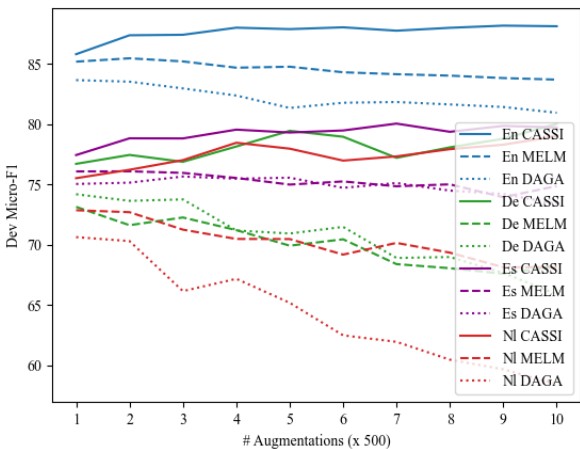

Figure 4: Dev set Micro-F1 vs Number of Augmentations for different methods on the 500 sentence subsets of CoNLL. Best viewed in color.

| Method | En 1k | De 1k | Es 1k | Nl 1k | Avg |
|---|---|---|---|---|---|
| Gold-Only | -1.02 ± 0.64 | -0.72 ± 0.51 | -0.63 ± 0.52 | -1.05 ± 0.65 | -0.855 ± 0.58 |
| Augs | -1.05 ± 0.70 | -0.71 ± 0.53 | -0.58 ± 0.48 | -1.11 ± 0.84 | -0.863 ± 0.64 |

Table 8: LM Scores on 1k gold sentences and 1k augmentations. LM scores are pseudo-log-likelihoods calculated using mBERT$_{\text{BASE}}$-cased following Salazar et al. 2020. The scores are normalized by the number of words in the sentence.

## 5 Conclusion

In this work, we propose CASSI as a text augmentation scheme for low-resource NER. CASSI produces contextually diverse sentences while preserving entity tags by structurally combining a pair of semantically similar sentences through the generation of candidate combinations that are filtered for semantic correctness and specificity. It leverages dependency parsers for structural information and a pretrained MLM for scoring. Through extensive experiments, we show that it outperforms baseline methods for multiple languages in monolingual and cross-lingual settings. Furthermore, we show that it is also able to outperform baseline methods under noisy text conditions where dependency parsing is likely to be unreliable.

## Limitations

The proposed method requires a dependency parser in the target language which might not be available for some extremely low-resource languages. Currently, Universal Dependencies contains treebanks for 141 languages[5] (Nivre et al., 2020) and SpaCy only supports 24 languages.

---

[5]https://universaldependencies.org/

For high-resource languages, increasing context diversity provides limited performance improvements for clean text regular NER outside of the low-resource setting.

## Acknowledgements

This research/project is supported by the National Research Foundation, Singapore under its AI Singapore Programme (AISG Award No: AISG2-GC-2022-005). We would like to acknowledge the High Performance Computing Centre of Nanyang Technological University Singapore, for providing the computing resources, facilities, and services that have contributed significantly to this work.

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

## A  Measuring Semantic Diversity of the Augmentations

We measure the semantic diversity of the augmentations when added to the gold sentences using Self-BERTScore, a diversity metric similar to Self-BLEU (Zhu et al., 2018)[6]. We score the 1k sets of CoNLL 2002/2003 using mBERT$_{BASE}$-cased in three cases: (i) Gold-Only, (ii) Gold-Only along with 5k augmentations filtered using the J-score filter, and (iii) Gold-Only along with 5k augmentations picked using the Best LM Score. Table 9 shows that augmentations selected using the J-Score metric are more diverse than those selected using the Best LM Score. (The intuition behind this

---

[6]Given the nature of our augmentations, the embedding-based BERTScore is a better metric than an n-gram based metric like BLEU. Furthermore, BERTScore outperforms BLEU at measuring diversity, though it still significantly underperforms on human evaluation (Tevet and Berant, 2021). Moreover, our augmentations essentially serve as an edge case for automated metrics since we do not formulate entirely new sentences and instead copy exact phrases from the source sentences.

| Set | En 1k | De 1k | Es 1k | Nl 1k |
|---|---|---|---|---|
| Gold-Only | *0.593 ± 0.057* | *0.622 ± 0.038* | *0.625 ± 0.055* | *0.621 ± 0.049* |
| J-Score | **0.599 ± 0.054** | **0.624 ± 0.034** | **0.633 ± 0.048** | **0.623 ± 0.042** |
| Best LM Score | 0.601 ± 0.054 | 0.629 ± 0.036 | 0.637 ± 0.048 | 0.625 ± 0.044 |

Table 9: Self-BERTScores calculated upon adding 5x augmentations to 1k gold-only sentences. Lower scores imply greater diversity.

may be clear from the following example: given two gold sentences $\mathbf{X}$ and $\mathbf{Y}$, 10 words long each with no overlap in words, and two candidate augmentations $\mathbf{C_1}$, which contains 5 words from $\mathbf{X}$ and 5 words from $\mathbf{Y}$, and $\mathbf{C_2}$, which contains 7 words from $\mathbf{X}$ and 3 words from $\mathbf{Y}$; intuitively, out of the two, $\mathbf{C_1}$ would be more diverse. This is reflected in the J-Scores: $\mathbf{C_1}$ has a J-Score of $0.333$, whereas $\mathbf{C_2}$ has a J-Score of $0.309$. Moreover, if the sentences in the original dataset are already well formed - which is the case in CoNLL - LM Scoring is likely to rank $\mathbf{C_2}$ over $\mathbf{C_1}$.)

Note that the Self-BERTScores of the augmentations are marginally higher than the gold-only sentences. This is expected for two reasons: (1) we pick the most semantically similar sentences for our augmentations, and (2) our augmentations consist of parts of the gold sentences used directly, i.e., the same words in the same order.

## B  Alternatives to Pairwise BERTScore

While pairwise BERTScore (calculation of BERTScore for every pair of sentences in the dataset) is efficient enough for small datasets, we recognize that using pairwise BERTScore becomes time-consuming for larger datasets since it involves token-wise operations for every pair of tokens in the two sentences. For this reason, we suggest the following alternatives for larger datasets: (1) pairwise cosine similarity using sentence embeddings (Reimers and Gurevych, 2019), and (2) randomly sampling a fixed number of sentences for each source sentence, and selecting the most semantically similar sentences from these random samples using BERTScore or sentence similarity.

## C  Examples of Generated Augmentations

Table 10 shows some example augmentations taken from a single 1k subset of CoNLL 2003 English. The examples show sentence interpolation, instances where the augmentation process leads to simple short phrase or word replacement, and instances of grammatical incorrectness/incoherence

in the augmentations. (Note that in the examples of simple phrase/word replacements the augmentations can still introduce entities with different tags or introduce an entity in place of a non-entity.)

| Gold Sentence Pair | Augmentation |
|---|---|
| The European Commission said on Thursday it disagreed with German advice to consumers to shun British lamb until scientists determine whether mad cow disease can be transmitted to sheep. 

 Bonn has led efforts to protect public health after consumer confidence collapsed in March after a British report suggested humans could contract an illness similar to mad cow disease by eating contaminated beef. | The European Commission said on Thursday humans could contract an illness similar to mad cow disease by eating contaminated beef. |
| Opel AG together with General Motors came in second place with 49,269 registrations, 16.4 percent of the overall figure. 

 The BLOM Stock Index which covers both markets rose 0.04 percent to 903.09 and the LISPI index rose 0.02 percent to 81.58. | Opel AG together with General Motors rose 0.04 percent to 903.09 and the LISPI index rose 0.02 percent to 81.58. |
| Defence Minister Volker Ruehe said that German troops would stay on in Bosnia next year as part of an international force to ensure the establishment of peace, a newspaper reported on Saturday. 

 Iraqi President Saddam Hussein has told visiting Russian ultra-nationalist Vladimir Zhirinovsky that Baghdad wanted to maintain friendship and cooperation with Moscow, official Iraqi newspapers said on Thursday. | Defence Minister Volker Ruehe said that German troops would stay on in Bosnia next year as part of to maintain friendship and cooperation with Moscow a newspaper reported on Saturday. |
| They said Zhirinovsky told Saddam before he left Baghdad on Wednesday that his Liberal Democratic party and the Russian Duma (parliament) are calling for an immediate lifting of the embargo imposed on Iraq after its 1990 invasion of Kuwait. 

 Defence Minister Volker Ruehe said that German troops would stay on in Bosnia next year as part of an international force to ensure the establishment of peace, a newspaper reported on Saturday. | They said German troops told Saddam before he left Baghdad on Wednesday that his Liberal Democratic party and the Russian Duma (parliament) are calling for an immediate lifting of the embargo imposed on Iraq after its 1990 invasion of Kuwait. |
| Lebed, who has arranged a military truce with separatist rebels in the southern Russia region, was in Moscow this week seeking support for a deal on Chechnya's political status. 

 Last October he was invited to attend the referendum held on Iraq's presidency, which extended Saddam's term for seven more years. | Lebed, who has arranged a military truce with separatist rebels in the southern Russia region, was in Moscow this week seeking to attend the referendum held on Iraq's presidency, which extended Saddam's term for seven more years. |
| Maronite Patriarch Sfeir expressed sorrow over the violations in Sunday's elections. 

 Fears of an Israeli operation causes the redistribution of Syrian troops' locations in Lebanon. | Maronite Patriarch Sfeir expressed sorrow over the redistribution of Syrian troops' locations in Lebanon. |
| Africans seeking to renew or obtain work and residence rights say Prime Minister Alain Juppe's proposals are insufficient as hunger strike enters 49th day in Paris church and Wednesday rally attracts 8,000 sympathisers. 

 Foreign Minister I.K. Gujral was asked at a news conference if India's decision to block adoption of the accord in Geneva would lead to an arms race with Pakistan and China. | Africans seeking to renew or obtain work and residence rights was asked at a news conference if India's decision to block adoption of the accord in Geneva would lead to an arms race with Pakistan and China. |
| American world champion Gwen Torrence, the bronze medallist in Atlanta, was second in 11.00. 

 Hever Golf Rose (11-4), last year's Prix de l'Abbaye winner at Longchamp, finished third, a further one and a quarter lengths away with the 7-4 favourite Mind Games in fourth. | American world champion Gwen Torrence, the bronze medallist in Atlanta, was second in Longchamp. |
| European champions Juventus will face English league and cup double winners Manchester United in this season's European Champions' League. 

 Ireland midfielder Roy Keane has signed a new four-year contract with English league and F.A. Cup champions Manchester United. | European champions Juventus will face a new four-year contract with English league and F.A. Cup champions Manchester United in this season's European Champions' League. |
| The Netherlands government has ruled out paying ransom money for a Dutch couple kidnapped from their farm, while Costa Rican authorities said on Tuesday they had no leads in the case. 

 Advanced Medical said it expects to take an unspecified one-time charge to pay for the merger. | The Netherlands government has ruled out it expects to take an unspecified one-time charge to pay for the merger, while Costa Rican authorities said on Tuesday they had no leads in the case. |

Table 10: Example augmentations generated from a 1k subset of CoNLL 2003 English. Instances of grammatical incorrectness/incoherence are underlined in red.