# OpenReview forum: "CASSI: Contextual and Semantic Structure-based Interpolation Augmentation for Low-Resource NER"
_EMNLP/2023/Conference — EMNLP 2023 Findings_

### Official Review · Reviewer_sAiY · 2023-08-01

**Soundness:** 4

**Excitement:**

4: Strong: This paper deepens the understanding of some phenomenon or lowers the barriers to an existing research direction.

**Paper Topic And Main Contributions:**

The authors of this paper proposes a new text augmentation technique aimed at addressing the challenges posed by low-resource Named Entity Recognition (NER) tasks. he authors introduce a novel approach (CASSI, Contextual and Semantic Structure-based Interpolation) which centers on the augmentation of NER-tagged text using dependency parsing-based exchange in pairs of semantically similar sentences. A key advantage of their method is its ability to generate fluent, contextually diverse, and semantically coherent text without corrupting annotations. The paper compares CASSI with two similar techniques (DAGA (Ding et al., 2020)) and MELM (Zhou et al., 2022)) in both low-resource monolingual and cross-lingual NER settings, demonstrating its superior effectiveness and improved performance compared to previous works.

**Reasons To Accept:**

- A novel technique that aims to improve the performance of NER systems in low-resource settings.
- The technique is well-explained and backed by a solid experimental setup that includes numerous experiments in various settings, including monolingual and cross-lingual settings.
- Section 4 offers interesting insights, including a more in-depth discussion about the results on a specific language and an ablation study that goes through various hyper-parameters of the text augmentation pipeline.
- Since the authors have included the code as supplementary material, I assume that it will be open-sourced.


**Reasons To Reject:**

None

**Reproducibility:**

4: Could mostly reproduce the results, but there may be some variation because of sample variance or minor variations in their interpretation of the protocol or method.

**Reviewer Confidence:**

3: Pretty sure, but there's a chance I missed something. Although I have a good feel for this area in general, I did not carefully check the paper's details, e.g., the math, experimental design, or novelty.

**Typos Grammar Style And Presentation Improvements:**

In Section 3.2.1, line 339, you say "[...] we set k to 5 [...]". I can assume that k is the top-k most similar sentences based on the LM-scoring. Am I right? I suggest to better specify what k is here, I feel like the reference to previous paragraph might be too weak.

---

> ### Author Rebuttal · Authors · 2023-08-29
>
> Thank you for your favourable review of our work.
>
> >_(The following is the response to "Presentation Improvement - In Section 3.2.1, line 339, you say "[...] we set k to 5 [...]". I can assume that k is the top-k most similar sentences based on the LM-scoring. Am I right? I suggest to better specify what k is here, I feel like the reference to previous paragraph might be too weak.")_
>
> You are correct, the $k$ on line 339 does refer to the top-$k$ sentence selection after LM Scoring. We will revise Section 3.2.1 to clarify this. Thank you for pointing this out.

---

### Official Review · Reviewer_vXhU · 2023-08-04

**Soundness:** 4

**Excitement:**

3: Ambivalent: It has merits (e.g., it reports state-of-the-art results, the idea is nice), but there are key weaknesses (e.g., it describes incremental work), and it can significantly benefit from another round of revision. However, I won't object to accepting it if my co-reviewers champion it.

**Paper Topic And Main Contributions:**

This paper proposes a data augmentation method for low-resource NER. The proposed method first generates candidate augmentations based on exchanging subtrees. The subtrees are predicted using an external dependency parser. Then, the generated candidates are filtered using (i) LM scoring or (ii) diversity and specificity creterions.

**Reasons To Accept:**

1. The motivation to augment data based on dependency parse tree is well-founded and interesting.
2. The proposed method outperforms previous data augmentation methods in monolingual NER.

**Reasons To Reject:**

It would be beneficial if the authors could demonstrate the effectiveness of the method on few-shot cross-domain NER, rather than only conducting experiments on cross-lingual settings, which are more common.

**Reproducibility:**

4: Could mostly reproduce the results, but there may be some variation because of sample variance or minor variations in their interpretation of the protocol or method.

**Reviewer Confidence:**

3: Pretty sure, but there's a chance I missed something. Although I have a good feel for this area in general, I did not carefully check the paper's details, e.g., the math, experimental design, or novelty.

---

> ### Author Rebuttal · Authors · 2023-08-29
>
> Thank you for your favourable review of our work.
>
> >_(The following is our response to "Reasons to Reject - It would be beneficial if the authors could demonstrate the effectiveness of the method on few-shot cross-domain NER, rather than only conducting experiments on cross-lingual settings, which are more common.")_
>
> While we agree that evaluation on cross-domain NER would be interesting, it is currently not feasible due to the lack of availability of cross-domain datasets that share tag-sets. Recent works on cross-domain NER [1-5] use cross-domain datasets with mismatched tag-sets, and attempt to design methods that address the issue of mismatching tag-sets. However, this would be out of the scope of this work. On the other hand, we evaluate on cross-lingual NER since it is more feasible because all 4 languages come from the CoNLL datasets and thus, share the same tag-set.
>
>
>
>
> ## References
>
> [1] Hu, Jinpeng, DanDan Guo, Yang Liu, Zhuo Li, Zhihong Chen, Xiang Wan, and      Tsung-Hui Chang. 2023. “A Simple Yet Effective Subsequence-Enhanced Approach for Cross-Domain NER”. Proceedings of the AAAI Conference on Artificial Intelligence 37 (11):12890-98.
>
> [2] Hu, Jinpeng, He Zhao, Dan Guo, Xiang Wan and Tsung-Hui Chang. 2022. “A Label-Aware Autoregressive Framework for Cross-Domain NER.” NAACL-HLT.
>
> [3] Jia, Chen and Yue Zhang. 2020. “Multi-Cell Compositional LSTM for NER Domain Adaptation.” Annual Meeting of the Association for Computational Linguistics.
>
> [4] Jia, Chen, Liang Xiao and Yue Zhang. 2019. “Cross-Domain NER using Cross-Domain Language Modeling.” Annual Meeting of the Association for Computational Linguistics
>
> [5] Lin, Bill Yuchen and Wei Lu. 2018. “Neural Adaptation Layers for Cross-domain Named Entity Recognition.” Conference on Empirical Methods in Natural Language Processing.

---

### Official Review · Reviewer_JhLS · 2023-08-11

**Soundness:** 4

**Excitement:**

3: Ambivalent: It has merits (e.g., it reports state-of-the-art results, the idea is nice), but there are key weaknesses (e.g., it describes incremental work), and it can significantly benefit from another round of revision. However, I won't object to accepting it if my co-reviewers champion it.

**Missing References:**

Regarding the statement in line 047, these are not necessarily “missing references” but are examples of augmentations that do not preserve the original label. SSMix represents an example of an augmentation that also aims to enhance contextual diversity like CASSI, though the result is not as natural.

- Soyoung Yoon, Gyuwan Kim, and Kyumin Park. 2021. SSMix: Saliency-Based Span Mixup for Text Classification. In Findings of the Association for Computational Linguistics: ACL-IJCNLP 2021, pages 3225–3234, Online. Association for Computational Linguistics.
- Fabrice Harel-Canada, Muhammad Ali Gulzar, Nanyun Peng, and Miryung Kim. 2022. Sibylvariant Transformations for Robust Text Classification. In Findings of the Association for Computational Linguistics: ACL 2022, pages 1771–1788, Dublin, Ireland. Association for Computational Linguistics.
- Qi Liu, Matt Kusner, and Phil Blunsom. 2021. Counterfactual Data Augmentation for Neural Machine Translation. In Proceedings of the 2021 Conference of the North American Chapter of the Association for Computational Linguistics: Human Language Technologies, pages 187–197, Online. Association for Computational Linguistics.

**Paper Topic And Main Contributions:**

This paper proposes an augmentation scheme for NER tasks that takes the following steps. Given a particular source text, (1) search the remainder of the dataset for the N most similar texts, (2) extract dependency parses from the selected texts and construct all combinations of subject and object/complement subtrees swaps, (3) filter the candidates via LM score for “correctness” / naturalness, (4) further filter the most natural candidates by Jaccard similarity with the source texts, which simultaneously increase specificity and offsets the genericizing effect of LM score filtering, and finally (5) some post-processing to fix potential IOB issues. Experiments on several mono- and cross-lingual NER datasets show the benefits of CASSI, especially for the lower resource settings and for languages like Dutch with lower representation in pretraining datasets. To highlight that CASSI is robust to imperfect dependency parses, the authors also explore augmentation on noisier social media datasets where CASSI still outperforms existing baselines.

Contributions include a new, high-performing augmentation pipeline that generates contextually diverse texts suitable for NER tasks as well as appearing general enough to be broadly applicable in other NLP task settings. Different elements of that pipeline, like LM scoring and Jaccard similarity filtering, can be adopted by other existing augmentation to improve the overall quality of the generations.

**Questions For The Authors:**

- Question A: Compared to the original sentences, how much lower/higher is the naturalness (LM score or other) of the augmented sentences on average? Relevant to justifying the “high quality” claim in line 011.
- Question B: Compared to the original sentences, how much lower/higher is the diversity (self-BLEU or other) of the augmented sentences on average? Relevant to justifying the “contextual diversity” claim in line 011. In general, one can intuitively see how the proposed approach increases diversity, but it would be helpful to quantify it for the community.
- Question C: What efficiency improvements have you considered for the various steps in CASSI? The time required scales as a function of dataset size due to the BERTScore similarity selection, which limits the feasibility of its use in larger datasets.

**Reasons To Accept:**

- The paper is well written and easy to follow. Frequently, when doubts would arise, they were often addressed in the next sentence or paragraph. Well done.
- CASSI explicitly considers both diversity and naturalness of the generations whereas many existing techniques yield ungrammatical, implausible, or incoherent texts. Some would argue that low-quality synthetic texts are acceptable because they still improve downstream model performance and generalization. However, in the real world, people often place significant weight on readability and having a technique that improves performance while being natural can help overcome objections from human decision makers and increase the likelihood of use.
- The evaluation setup acknowledges the limitations of the augmentation to low resource settings and sufficiently demonstrates its benefits relative to existing baselines.

**Reasons To Reject:**

- In general, the claims of “high-quality” and increased “contextual diversity” are left unquantified, at least directly. The high-quality claim may potentially be substantiated via improvements to model performance but given the gist of the filtering criteria suggests that a quality == naturalness argument is being made. Authors could use an approach like cleanlab (https://github.com/cleanlab/cleanlab) to support the claim that CASSI does not introduce any new label issues brought on by shifts in augmented semantics.

**Reproducibility:**

5: Could easily reproduce the results.

**Reviewer Confidence:**

4: Quite sure. I tried to check the important points carefully. It's unlikely, though conceivable, that I missed something that should affect my ratings.

**Typos Grammar Style And Presentation Improvements:**

Typos:
- Line 343 - is the Base supposed to be subscripted like that?

Presentation:
- Figure 1 - a missed opportunity to show your approach more clearly using a natural language example
- Line 047 - not all augmentations preserve the label, e.g. mixup augmentations, sibylvariant transforms, and counterfactual augmentations all alter semantics and derive new targets for the augmented data. See “missing references” for example works. Can clarify via “... while generally preserving the labels.”

Difficult to Follow:
- Line 276 - Unclear how filtering with Jaccard similarity increases diversity if you’re trying to find similar sentences. This filtering seems to merely offset the genericization of LM-scoring in the prior step and actually discourages diversity by keeping the augmentations that are more similar to existing samples in the dataset.

---

> ### Author Rebuttal · Authors · 2023-08-29
>
> Thank you for your favourable review of our work. We appreciate your detailed assessment and based on your comments we have decided to make the following additions to the paper. We discuss these in greater detail below.
> 1. Include the scores with regards to Question A and Question B in Section 4 (Discussion) of the paper
> 2. Clarify how J-Score filtering improves diversity
> 3. Indicate that not all augmentations preserve labels in the Introduction
> 4. Include a discussion of efficient alternatives to BERTScore in the Appendix
> 5. If space allows, change Figure 1 to include a natural language example (space constraints did not allow us to do so earlier)
>
> ## Quantifying “High Quality”
> >_(Includes responses to Question A, Question B, Reasons to Reject, and “Presentation Improvements - Difficult to Follow”)_
>
> While naturalness is a key part of our “high quality augmentations” claim, our claim  of “high quality” encompasses label preservation, context diversity, and naturalness/semantic correctness. In the paper, apart from the main results and ablations, we attempt to support the high quality claim via the number of augmentations (Section 4.3) in two ways: (i) by showing that we are able to add significantly more augmentations than baseline methods, and (ii) by showing that our method outperforms baselines at all augmentation levels. However, we agree that separately addressing the different properties of our augmentations would be very helpful. We discuss this further below:
>
> ### 1. Naturalness
> >_(Response to Question A -  Compared to the original sentences, how much lower/higher is the naturalness (LM score or other) of the augmented sentences on average? Relevant to justifying the “high quality” claim in line 011.)_
>
> We performed LM scoring on the 1k subsets of CoNLL 2002/2003, and 1k augmentations generated from them. We scored the sentences using `mBERT-Base-cased` (we do not use `XLM-R-Base` since the augmentations were selected using `XLM-R-Base`) following Salazar et al. [1] and normalized them by the number of words (ignoring punctuation). The augmentations are scored separately, i.e, they are not added to the gold-only set. The LM Scores are log-likelihoods, with greater values being better. We present the mean and standard deviation of the LM scores over the set. The table below shows that the augmentations score very similarly (albeit marginally lower on average) to the gold sentences.
>
> | Method    | En         | De         | Es         | Nl         | Avg         |
> |-----------|------------|------------|------------|------------|-------------|
> | Gold-Only | -1.02 ± 0.64 | -0.72 ± 0.51 | -0.63 ± 0.52 | -1.05 ± 0.65 | -0.855 ± 0.58 |
> | Augs      | -1.05 ± 0.70 | -0.71 ± 0.53 | -0.58 ± 0.48 | -1.11 ± 0.84 | -0.863 ± 0.64 |
>
>
> ### 2. Context Diversity
> >_(Includes responses to:_
> >- _Question B - Compared to the original sentences, how much lower/higher is the diversity (self-BLEU or other) of the augmented sentences on average? Relevant to justifying the “contextual diversity” claim in line 011. In general, one can intuitively see how the proposed approach increases diversity, but it would be helpful to quantify it for the community._
> >- _“Difficult to Follow; Line 276” - Unclear how filtering with Jaccard similarity increases diversity if you’re trying to find similar sentences. This filtering seems to merely offset the genericization of LM-scoring in the prior step and actually discourages diversity by keeping the augmentations that are more similar to existing samples in the dataset.)_
>
> Context Diversity is difficult to directly quantify in the case of NER, since it has to do with entities in new settings, to help the model find entities in unseen examples. The new settings have to be semantically consistent with the entity. Therefore, two sentences can have different contexts (with respect to entities) but similar semantics. This is certainly the case with our augmentations because we select the most semantically similar sentences. Therefore, to be reliably measured, context diversity needs to be decoupled from the semantic “topic” diversity of the sentences. Most sentence similarity/diversity metrics like BLEU, BERTScore, etc., are inherently biased towards measuring semantic similarity. Nonetheless, we scored our augmentations and the gold dataset, as shown below, using “Self-BERTScore”, calculated in a similar way as Self-BLEU (BLEU was not a suitable metric given the nature of our augmentations, since BLEU scores via matching n-grams, which leads to  5x - 6x higher scores when gold-only and augmentations are combined. BERTScore is more suitable since it measures cosine similarities between word embeddings.)
>
> | Set                                 | En                | De                | Es                | Nl                |
> |-------------------------------------|-------------------|-------------------|-------------------|-------------------|
> | Gold-Only (1k)                      | **0.593 ± 0.057** | **0.622 ± 0.038** | **0.625 ± 0.055** | **0.621 ± 0.049** |
> | _Gold-Only (1k) + J-Score (5k)_     | _0.599 ± 0.054_   | _0.624 ± 0.034_   | _0.633 ± 0.048_   | _0.623 ± 0.042_   |
> | Gold-Only (1k) + Best LM Score (5k) | 0.601 ± 0.054     | 0.629 ± 0.036     | 0.637 ± 0.048     | 0.625 ± 0.044     |
>
>
> We score the 1k sets of CoNLL 2002/2003 using mBERT-Base-cased in three cases: (i) Gold-Only, (ii) Gold-Only along with 5k augmentations filtered using the J-score filter, and (iii) Gold-Only along with 5k augmentations picked using the Best LM Score (to address “Difficult to Follow; Line 276”).
>
> Self-BERTScore is scored between -1 and 1, with lower scores indicating higher diversity. Along with the Self-BERTScore, we also include the standard deviation of the calculated BERTScores in each case.
>
> The table above shows that the augmentation largely maintains the semantic diversity in the dataset. While the augmentation diversity scores are marginally higher, they fall well within the first standard deviation of the gold-only scores.
>
> Furthermore, the table also shows that using J-Score filtering does consistently result in higher diversity as compared to using the Best LM Score. (Relevant to “Presentation Improvements: Difficult to Follow - Line 276”. The intuition is that given two gold sentences $X$ and $Y$, $10$ words long each with no overlap in words, and two candidate augmentations $C_1$, which contains $5$ words from $X$ and $5$ words from $Y$,  and $C_2$, which contains $7$ words from $X$ and $3$ words from $Y$; intuitively, out of the two, $C_1$ would be more diverse. This is reflected in the J-Scores: $C_1$ has a J-Score of $0.333$, whereas $C_2$ has a J-Score of $0.309$. Moreover, if the sentences in the original dataset are already well formed - which is the case in CoNLL - LM Scoring is likely to rank $C_2$ over $C_1$)
>
>
> ### 3. Label Preservation
> >_(Response to second half of “Reasons to Reject” - Authors could use an approach like cleanlab to support the claim that CASSI does not introduce any new label issues brought on by shifts in augmented semantics.)_
>
> As suggested, we used CleanLab to find potential label issues in the augmentations. CleanLab inputs a datasets labels and prediction probabilities over the labels for each instance, and outputs outliers based on incorrect predictions with high confidence. We performed inference using the gold-only 1k English set on 5k augmentations generated from the set (we only used English to be able to manually detect semantic shifts). For incorrect classifications from the gold-only model, upon manual inspection, we found no incorrect labelling explicitly introduced by the augmentations. Specifically, almost all cases of incorrect classifications were that the entities had been classified as “I-MISC”, which is the most underrepresented entity class in the dataset. The remaining label issues originated from incorrect labelling in the original training set.
>
>
> ## Alternatives to BERTScore
> >_(Response to Question C -  What efficiency improvements have you considered for the various steps in CASSI? The time required scales as a function of dataset size due to the BERTScore similarity selection, which limits the feasibility of its use in larger datasets.)_
>
> While pairwise BERTScore (calculation of BERTScore for every pair of sentences in the dataset) is efficient enough for the sizes of the low-resource datasets we work with in the paper, we recognize that using pairwise BERTScore becomes time consuming for larger datasets (taking 6 hours for a 15k sentence dataset on an Nvidia PV100 PCIe GPU). We suggest considering the following alternatives to the pairwise BERTScore calculation:
> - Cosine Similarity using Sentence Embeddings
> - For larger datasets, a constant number of sentences can be randomly sampled for each source sentence, and the most semantically similar sentence(s) can be selected from these random samples using BERTScore/Sentence Similarity.
>
>
> ## Conclusion
>
> This concludes our response. As mentioned in the beginning, we will revise the paper to accommodate your suggestions. We believe this will help improve the quality of our paper. Once again, thank you for all your valuable feedback on our work!
>
> ## References
>
> [1] Salazar, Julian, Davis Liang, Toan Q. Nguyen and Katrin Kirchhoff. 2019. “Masked Language Model Scoring.” Annual Meeting of the Association for Computational Linguistics.

---

### Meta-Review · Area_Chair_JUwu · 2023-09-19

**Recommendation:** 4

**Metareview:**

The paper introduces an NER augmentation method, which involves finding similar texts in the dataset, generating various subtree swaps, and filtering candidates based on language model scores and Jaccard similarity. Experiments demonstrate its effectiveness, particularly in low-resource settings and for languages like Dutch. The method remains robust even with imperfect dependency parses, outperforming existing baselines on noisy social media datasets. Notwithstanding the inquiries or apprehensions raised by the reviewers, the authors adeptly resolved them in their rebuttal.

---

### Decision · Program_Chairs · 2023-10-07

**Decision:**

Accept-Findings

**Comment:**

The paper introduces an NER augmentation method, which involves finding similar texts in the dataset, generating various subtree swaps, and filtering candidates based on language model scores and Jaccard similarity. Experiments demonstrate its effectiveness, particularly in low-resource settings and for languages like Dutch. The method remains robust even with imperfect dependency parses, outperforming existing baselines on noisy social media datasets. Notwithstanding the inquiries or apprehensions raised by the reviewers, the authors adeptly resolved them in their rebuttal.